# Model Parametrization-Based Genetic Algorithms Using Velocity Signal and Steady State of the Dynamic Response of a Motor

**DOI:** 10.3390/biomimetics10030146

**Published:** 2025-02-27

**Authors:** Mayra Cruz-Fernández, J. T. López-Maldonado, Omar Rodriguez-Abreo, Alondra Anahí Ortiz Verdín, J. Iván Amezcua Tinajero, Idalberto Macías-Socarrás, Juvenal Rodríguez-Reséndiz

**Affiliations:** 1Division de Tecnologías Industriales, Universidad Politécnica de Querétaro, Santiago de Querétaro 76240, Mexico; mayra.cruz@upq.edu.mx (M.C.-F.); alondra.ortiz@upq.mx (A.A.O.V.); jose.amezcua@upq.edu.mx (J.I.A.T.); 2Red de Investigación OAC Optimización, Automatización y Control, El Marques 76240, Mexico; 3Escuela de Ingeniería, Universidad Anahuac de Querétaro, Querétaro 76246, Mexico; juvenal@uaq.edu.mx; 4Facultad de Ciencias Agrarias, Universidad Estatal Península de Santa Elena, Santa Elena (UPSE), Libertad 240204, Ecuador; imacias@upse.edu.ec

**Keywords:** genetic algorithms, dynamic model, aritficial inteligence, parameter estimation

## Abstract

The study of dynamic models and their parameterization remains a relevant topic in research. Motors and their models have been extensively analyzed, studied, and parameterized using various techniques due to their broad applicability in motorering and industrial settings. However, most methods for obtaining model parameters require at least two averaged signals from the motor, such as torque, current, speed, position, or acceleration. In this work, we propose the parameterization of a motor’s dynamic model using only the speed signal and the steady-state values of the variables. Through evolutionary computation, the mechanical and electrical equations of the motor are reconstructed based on this signal. This approach offers a significant advantage, as it enables parameter estimation without requiring the instrumentation needed for full current signal measurement or, alternatively, torque measurement. To achieve this, the transfer function representing the motor’s speed is utilized. The function reconstruction is performed with a Root Mean Square Error (RMSE) of less than 1% for both the speed and current signals. Since the original current signal is not required for this estimation, this work presents an innovative approach to estimating a system of dynamic equations using only a single measured variable and the dynamic relationships of its step-input response.

## 1. Introduction

Parameter identification and controller tuning are essential challenges in motorering. In particular, permanent magnet direct current (DC) motors [1] have gained significant importance due to their growing utilization in high-demand applications.

DC motors are a fundamental component in the contemporary industrial and technological landscapes. Their inherent simplicity in construction, coupled with the capability to deliver precise control over speed [2] and torque, positions them as an optimal solution for various applications, spanning fields such as robotics [3], system optimization, and industrial automation. As efforts are made to penetrate markets ranging from compact medical devices to expansive propulsion systems in electric vehicles, these motors play a critical role in addressing the sophisticated control requirements of increasingly challenging environments [4]. However, the precise identification of motor parameters and the formulation of models that effectively encapsulate their true dynamics continue to pose significant challenges, such as the technological development in the design of environmentally friendly vehicles [5], or the design of a controller based on the PID control method with MATLAB [6]. Another important motorering application that often poses a challenge in conjunction with the boom in the automotive industry and the use of digital tools is automotive predictive maintenance, which has been evolving in terms of identifying the parameters under consideration with the help of artificial intelligence (AI) and the application of stochastic methods [7]. This difficulty is largely attributed to the complex interaction of external factors, such as load variations, environmental conditions and mechanical degradation [8].

Advancements in technology have elevated the requirements for direct current (DC) motors, necessitating enhanced efficiency and adaptability in their control mechanisms. Conventional models that rely on linear approximations and simplifications often prove inadequate in capturing the nonlinear behaviors and complex dynamics inherent to these systems [9]. Consequently, there is a pressing need to devise innovative strategies that not only enable precise modeling of these motors, but also optimize their performance for particular applications.

An essential component in the application of these motors is their tuning and control. This interest has stimulated numerous investigations aimed at modifying dynamic parameters and creating controllers that enhance their performance. Among control strategies, proportional integral derivative (PID) controllers are particularly notable for their simplicity, robustness, and widespread adoption within the industry [10]. However, the precise adjustment of these controllers continues to pose significant challenges, particularly when traditional methods such as the Ziegler-Nichols method are applied [11]. Although these conventional approaches are widely utilized, they often fall short of delivering optimal performance due to their inherent limitations in accommodating systems with complex charging dynamics. To address these shortcomings, multiple artificial intelligence (AI) and evolutionary computation techniques have been applied to the parameterization problem. AI has influenced various fields of knowledge, ranging from economics [12] and biology [13] to motorering [14]. Evolutionary computation, a branch of AI, includes metaheuristic algorithms that have demonstrated exceptional performance in optimization and search problems, such as genetic algorithms (GA) [15] and Cuckoo Search [16]. These methodologies have proven to be highly effective in identifying parameters that ensure a stable, fast, and precise dynamic response, tailored to meet the specific requirements of various applications [17].

Numerous studies have concentrated on enhancing the performance of these motors using approaches such as compensation for commutation errors and precise estimation of position and angular velocity [18]. Furthermore, some researchers are seeking to identify these parameters while the motor operates (online) to analyze the variation of parameters under varying operating conditions, such as load fluctuations and speed, utilizing a particle swarm optimization (PSO) algorithm [19]. For online identification of control parameters, sophisticated techniques, including orthogonal function-based control [20], have been developed to improve the accuracy of the motor dynamic model. A major challenge in this context is the parametric estimation of the motors due to the uncertainty of their parameters. This approach is validated through simulations and experimental tests, demonstrating that the proposed algorithm offers higher accuracy than traditional methods [21], presenting an innovative, adaptable, and effective methodology for the tuning of PID controllers in applications necessitating dynamic and adaptive adjustments, with the possibility of modifying the speed [22] and accuracy of the system according to the particular needs of each assignment.

The exclusive use of the velocity signal in this study to develop dynamic model parameters for a DC motor constitutes a noteworthy advancement in the realms of motor modeling and control, especially in environments with limited resources. Traditional methods of modeling DC motors often require multiple measurements, including voltage, current, and various mechanical parameters to accurately represent the motor’s behavior. This can be a limitation in situations where measurement tools are scarce or where the complexity of the system demands a more streamlined approach. A comparison between works is shown in Table 1 where it is shown that typically two variables are used for the parametric estimation of an motor.

Considering the above, the main contributions of this work are:Using a single signal for parameterization of the dynamic model of an motor.Leveraging stationary relationships for single-signal parametric estimation.Adaptability to any metaheuristic algorithm.

By focusing on a single variable, the velocity signal, this methodology simplifies the modeling requirements. The speed signal is a key parameter for understanding the dynamic performance of a DC motor, as it directly reflects the motor’s operational state. The relationship between speed and the motor’s mechanical and electrical equations can be effectively leveraged to develop a model.

The remainder of this paper is organized as follows: Section 2 describes the methodology employed in this study, including the mathematical modeling of the motor and the genetic algorithm used for parameter estimation. Section 3 presents and discusses the results, comparing the original signals with the reconstructed signals obtained from the estimated parameters. Finally, Section 4 provides the conclusions of this research.

## 2. Materials and Methods

A direct current motor is an electromechanical system. In simple terms, it can be described as a transducer of electrical energy to mechanical energy; a simple diagram of the direct current motor is shown in Figure 1.

In this work, the motor parameters are estimated in stages. The first stage consists of estimating three coefficients of the speed Transfer Function (TF) using a Genetic Algorithm (GA). Subsequently, the steady-state equations are used to calculate the final parameter of the TF. Finally, after the iterative procces, the remaining dynamic model parameters are determined. The complete process is illustrated in Figure 2.

### 2.1. Dynamic Description of a Direct Current Motor

The dynamic model of a direct current motor is widely known and has been used in several recent works [27,28]. Its general model consists of 4 equations that relate the mechanical and electrical parts of this hybrid system (see Equation (Equation 1)).(1)$$\left.\begin{array}{c} V\left(t\right)=RI\left(t\right)+L\frac{dI(t)}{dt}+E\left(t\right)\\ \tau \left(t\right)=J\frac{d\omega (t)}{dt}+B\omega \left(t\right)+T_{L}\\ E\left(t\right)=K_{e}\omega \left(t\right)\\ \tau \left(t\right)=K_{m}I\left(t\right) \end{array}\right\}$$
where v(t) is the direct current voltage signal applied to the motor; R is the resistance of the motor winding; J is the moment of inertia of the rotor; B is the coefficient of friction; L represents the value of the inductance; E is the voltage induced in the armature; τ is the motor torque; I(t) is the current consumed by the motor; ω is the angular velocity of the rotor; $T_{L}$ is the torque that the motor generates to move external loads; $K_{e}$ is the electrical constant; $K_{m}$ is the mechanical constant.

Since this is a parametrization problem, the tests were performed with a constant voltage, that is, V(t) is a step type input. The step response is widely known and is usually composed of two parts, the transient part where the system shows highly variable behavior and the stationary part where the system response remains constant. An example of these parts is shown in Figure 3.

As additional considerations, null initial conditions are typically assumed in parametrization problems. Additionally, $T_{L}$ is set to zero, as step response tests are usually performed with a load-free motor. It can also be assumed that the electrical constant is of a similar magnitude to the mechanical constant; therefore, a single constant K can be defined such that $K=k_{e}=k_{m}$, as stated in [29]. By performing algebraic manipulations, the system of equations in (Equation 1) can be reduced to the simplified system shown in (Equation 2).(2)$$\left.\begin{array}{c} v\left(t\right)=RI\left(t\right)+L\frac{dI(t)}{dt}+K\omega \left(t\right)\\ KI\left(t\right)=J\frac{d\omega (t)}{dt}+B\omega \left(t\right) \end{array}\right\}$$

An alternative to study the system of equations of a linear dynamic system is to use the transfer function which allows to relate the output to the input in the Laplace space. Using the Laplace transformation in the system of Equation (Equation 2) and with the corresponding algebraic operations we can obtain the transfer functions shown in Equation (Equation 3) for the velocity signal and the transfer function of Equation (Equation 4) for the current signal.(3)$$\frac{\omega (s)}{V(S)}=\frac{K}{LJS^{2}+\left(RJ+LB\right)S+\left(RB+K^{2}\right)},$$(4)$$\frac{I(s)}{V(S)}=\frac{JS+B}{LJS^{2}+\left(RJ+LB\right)S+\left(RB+K^{2}\right)},$$

The parameterization problem of a DC motor is typically analyzed by comparing real signals of current and speed with those estimated by various methods. Some studies replace the current signal with the torque signal, while others use the acceleration or position signal instead of velocity. However, the present work focuses on solving the problem without relying on the complete current signal. Only the steady-state current value is assumed to be known. As observed in the Equation (Equation 3), this does not depend on the current and can be formulated as a parametric estimation problem of a second-order transfer function with the following structure:(5)$$\frac{\omega (s)}{V(S)}=\frac{a}{bS^{2}+cS+d},$$

Previous research has shown that second-order and higher systems can be described by second-order transfer functions [30]. This allows reducing the number of parameters to be searched for to *a*, *b*, *c* and *d* parameters. Aditionally, the following relationships can be observed:(6)a=k,(7)b=LJ,(8)c=RJ+LB(9)$$d=RB+K^{2},$$

Considering the above, the problem becomes the parametrization of the transfer function of velocity. This problem can be solved with various methods; however, greater control of the search is obtained and a highly flexible implementation with metaheuristic methods [31]. In adition, there are the steady-state relationships of the direct current motor which are described by Equations (Equation 10) and (Equation 11) as described in the research [32].(10)$$B=\frac{KI_{ss}}{\omega_{ss}},$$(11)$$R=\frac{V_{ss}-K\omega_{ss}}{I_{ss}},$$
where $\omega_{ss}$, $I_{ss}$ and $V_{ss}$ represent the steady-state velocity, steady-state current, and steady-state applied voltage, respectively. Measuring the full current signal may require complex instrumentation, depending on the level of noise present, or, alternatively, more expensive sensors. However, obtaining the steady-state current is a straightforward procedure that can be carried out using any ammeter. Therefore, if the steady-state values of the variables are known, the parameters B and R in Equations (Equation 10) and (Equation 11) can be calculated, respectively, if we know the value of the parameter *K*. Considering Equation (Equation 6), the metaheuristic algorithm, when proposing a value for *a*, also generates a candidate solution for *k*. Consequently, the candidate coefficients *B* and *R* can be calculated using Equations (Equation 10) and (Equation 11). Finally, with the candidate values of *k*, *B*, and *R*, the parameter *d* can be determined using Equation (Equation 9). This approach reduces the problem to searching for only three parameters of the transfer function while ensuring that the steady-state relationships are satisfied.

### 2.2. Simulation of the Dynamic Model of the DC Motor

Two motors with known parameters were used to test the proposal of this work: the Mavilor MLC050 (Mavilor, Barcelona, Spain) and the RMCS2004 (Robokits, Gujarat, India). The Mavilor 3000 rpm motor operates at a maximum voltage of 24 volts, with a nominal current consumption of 3.9 amps and a nominal power of 47 watts. It is equipped with a square wave encoder that generates 1000 pulses per revolution, powered at 5 volts, allowing precise speed measurement. On the other hand, the RMCs2004 motor, manufactured by Robokits, has a nominal voltage of 24 V, a nominal power of 100 watts, and a maximum speed of 6000 rpm. The acquisition of real parameters is performed by collecting current and voltage signals using a proprietary acquisition card based on Arduino, the integrated encoder, and a Hall effect sensor for current measurement. The sampling rate is set to one millisecond for three seconds. Both the encoder and the current sensor are sampled every millisecond for three seconds at a constant voltage of 10.5 volts. Once the signals are acquired, the nominal parameters are obtained using the Steiglitz–McBride method. A full description of the acquisition and process is provided in [33]. Additionally, steady-state and transient-state relationships have been utilized to facilitate parameter estimation with metaheuristic algorithms [34]. Table 2 shows the nominal values of both motors used in this work; for comparison purposes, these motors are the same as those used in the investigation [33].

Simulink was used to recreate the dynamic response of these DC motors. Within this environment, the system of Equations described by (Equation 3) and (Equation 4) is developed. The complete simulation is depicted in Figure 4.

In order to have similar sampling parameters to the previous works, the numerical method ODE3 with a fixed step with intervals of 0.001 s is used and each motor is simulated for 2.5 s. As a result of these simulations, complete responses to a step input for both motors are obtained. In addition, an additional simulation is also needed for the transfer function, for which the scheme shown in Figure 5 was used.

Finally, the program illustrated in Figure 5 was used by the genetic algorithm programmed in Matlab to estimate the parameters a, b, c, and d of the velocity transfer function.

### 2.3. Genetic Algorithms as a Parametric Estimator in Transfer Functions

Genetic algorithms (GAs) are the most widely used metaheuristic algorithms [35], and their effectiveness has been demonstrated in numerous applications [36], including parametric estimation [37]. They have even been successfully applied to the parametric estimation of DC motors [38]. However, previous research has used the entirety of both signals for parametric estimation, whereas the present work relies solely on the velocity signal and steady-state values.

These algorithms are inspired by the natural selection process, where the fittest individuals are chosen for reproduction. GAs operate on a population-based approach, applying evolutionary operators to converge toward a solution that meets the search requirements. There are a large number of variations in genetic algorithms, the code used in this work is based on the GA developed in [39]. A general flowchart of the employed genetic algorithm and its evolutionary operators is presented in Figure 6.

To construct the genetic algorithm, it is necessary to generate a random initial population. This population consists of a set of individuals, each representing a possible solution encoded in a chromosome, which in turn is composed of genes.

If Equation (Equation 6) is considered, it can be observed that k=a, therefore, with a proposed parameter K, a parameter B can be obtained that respects the steady-state relationship shown by Equation (Equation 10), in the same way the parameter R can be proposed by being given in Equation (Equation 11). Subsequently, parameter d can be calculated using Equation (Equation 9). This allows to reduce the search of the algorithm since from 3 parameters (a, b and c) parameter d is calculated that will respect the steady-state relationships.

Since the parameterization initially requires three values, the chromosome is represented as a vector containing three parameters: a, b, c which correspond to the parameters of the transfer function (Equation 3), as shown in Equation (Equation 12).(12)Chromosome=[abc],

Once the initial population is constructed, the parameter *d* is calculated, and the ability of each individual to solve the problem is evaluated. The parameters of the dynamic model are assessed using a cost function, commonly referred to as the fitness function. In this work, one of the most widely used metrics, the Root Mean Square Error (RMSE), is employed. The formula for RMSE is presented in Equation (Equation 13).(13)$$RMSE=\sqrt{\frac{\sum_{i=1}^{n}{\left(\hat{\omega }-\omega \right)}^{2}}{n}},$$
where $\hat{\omega }$ are the values estimated by the genetic algorithm, and n is the number of samples in the velocity signal. The fittest individuals are those with a lower RMSE; ideally, the perfect individual with an RMSE value of zero would be sought. A better fitness value increases the probability that individuals reproduce.

The selection of individuals for reproduction was performed using a roulette wheel mechanism. The GA implemented in this study incorporates elitism, ensuring that the best individuals are selected for reproduction. The fittest individuals are then combined using a single-point crossover method, where a random crossover point is chosen. This process ensures that each offspring inherits a portion of its genetic material from the first parent and the remaining part from the second parent. From each pair of parents, two offspring are generated: the first inherits the initial gene until the crossover point from parent 1 and the remaining segment from parent 2, while the second offspring follows the opposite pattern.

Once the offspring are generated, these new individuals may undergo a random mutation, meaning that one of their genes can change.The mutation was implemented by generating a random number. If the mutation hyperparameter is greater than the generated random number, one of the genes is modified to a new value within the search range. This mechanism helps the GA escape local optima during the search process. The procedure is repeated over multiple generations (iterations) until the desired error threshold is reached. The search process is regulated by tuning the hyperparameters, whose values used in this research are summarized in Table 3.

The upper limit search value of K is determined by Equation (Equation 11), since if K were greater than the proposed value it would produce a negative value in R which makes no physical sense. With the maximum value of K, a maximum possible value of B can also be obtained using the Equation (Equation 10). With the maximum value of these two parameters, the maximum value search of the parameters a and d can be determined; the rest of the limits are wider because there is no equation to estimate the search range. Once the genetic algorithm is configured with the hyperparameters in Table 3, the iterations are performed to achieve 200 generations.

Once the iterative process is completed, the parameters K, B and R are obtained. Thus, the problem is reduced to estimating *L* and *J*. By analyzing Equations (Equation 7) and (Equation 8), it can be observed that we have a two-by-two system of equations where the unknowns are the parameters *L* and *J*. Using algebra to combine these expressions, Equation (Equation 14) is derived.(14)$$-BL^{2}+Lb-Ra=0,$$

By solving the quadratic equation, two possible values for *L* are obtained, resulting in a pair of potential solutions. Considering the typical response of Equation, the *J* with the greater magnitude is selected.

## 3. Results and Discussion

This section presents the results of the parametric estimation of the dynamic dc motor model using only the speed signal and steady-state values. For comparison purposes, two motors with known nominal parameters were used. After running in MATLAB/Simulink environment the algorithm shown in Figure 6 with the hyperparameters listed in Table 2, the parameters of Equation (Equation 3) are obtained and results are depicted in Table 4.

The results reveal a percentage difference of 7.28% for parameter *a*, 1.0931% for parameter *b*, 1.108% for parameters *c* and, 1.3903% for *d* for Mavilor Motor. With these parameters, when simulating the velocity equation, the graph shown in Figure 7 is obtained. This model has an RMSE at velocity of 0.1578%.

The algorithm’s cost function is illustrated in Figure 8. The horizontal axis represents the number of fitness function evaluations, while the vertical axis shows the evolution of the RMSE magnitude. The number of iterations is not expressed directly because some metaheuristic algorithms such as cuckoo search [40] employ a double evaluation of the fitness function for each iteration.

As a second test motor, the genetic algorithm is executed with the same hyperparameters exhibited in Table 2, obtaining the results shown in Table 4. The estimated velocity compared to the simulated velocity is shown in Figure 9 and an RMSE of 0.2492% was obtained. Additionally, a percentage difference of 0.1008% for parameter *a*, 0.54% for parameter *b*, 0.26% for parameter *c*, and 0.09% for parameter *d* for RMCS2004 Motor. These results show that a minimal error in the estimation of coefficient a also considerably reduces the errors in the estimation of coefficients b, c, and d. In the case of the RMCS2004 motor, no parameter exceeds 1% error. However, both reconstructions of the speed signal show an RMSE of less than 1%.

The reduction of the cost for the RMSC2004 motor is depicted in Figure 10.

In both cost functions, is observed the reduction of the RMSE to values below 0.01. If greater precision is required, iterations can be continued, and although it is possible to further reduce the cost, the computational cost grows exponentially due to the increase in the problem size. This method involves numerical integration using Simulink, which represents one of the major disadvantages compared to other methods. Once the parametric estimation of the Transfer Function of the velocity is completed, the parameters of the dinamic model of motor are calculated using the steady-state relationships with Equations (Equation 10), (Equation 11) and (Equation 14). The results of the parametric estimation of the motors from the coefficients of the velocity transfer functions are shown in Table 5.

A K value error cuases a variation in B and R, since these parameters are calculated from the value of K. In turn, there is an accumulated error in the calculation of parameters J and L. These results indicate that the correct parametric estimate has a high dependence on the correct estimate of K, since errors in this parameter will invariably cause a variation in the other parameters of the dynamic model of the DC motor. Since the 5 parameters of the motor model have already been obtained, the complete current graph can be obtained.

The graph of the actual current of the mavilor motor and its comparison with the estimated current obtained with Simulink through the Equation (Equation 4) with the calculated parameters is shown in Figure 11.

On the other hand, the estimation of the current versus the real current in the RMSC2004 motor is shown in Figure 12.

The original current signal is assumed to be unknown so the signals shown in Figure 11 and Figure 12 are only shown for comparative purposes. From these graphs, it can be observed that the error is greater in the Mavilor motor; in comparison, the reconstruction of the signals of the RMCS motor had a considerably smaller error in magnitude. However, the RMSE in the reconstruction of the speed signal remains in similar magnitudes for both motors. The RMSE in the reconstruction of the current for the case of the Mavilor motor was 0.33%, while in the RMCS motor, a RMSE of 0.31% was found for the estimation of the current. In both cases, the RMSE found is less than 1%, which shows the validity of the proposed method.

The main limitations of this method have been addressed, clarifying that it is only an optional approach when both complete signals are unavailable and serves as an alternative for model parameterization. These limitations stem primarily from the constraints of metaheuristic algorithms, the most notable being the proper selection of hyperparameters. However, self-tuning methods can be implemented to mitigate this issue.

The main assumptions arise from the modeling stage, including the premise that both electrical and mechanical constants are of similar magnitude. Furthermore, the initial conditions are taken as zero, external load are zero too and a step response was used as input signal. Additionally, while this method introduces higher-order errors compared to other established techniques, it still provides an approximate representation that allows for the implementation of a control system accounting for parametric uncertainty.

Finally, the computational cost is higher than that of traditional methods, although it largely depends on the chosen hyperparameters.

The method was designed for DC motors; however, it would extended to other types of dynamic systems described by a set of ordinary differential equations, provided that their steady-state relationships are fully defined and some variables are difficult to measure.

## 4. Conclusions

In this work, a parametric estimation of a motor is performed using its speed transfer function and the steady-state values of the variables. This method enables the calculation of the motor’s dynamic model and the transfer functions for both current and velocity variables. Unlike other methods, it does not require the current signal for parameter estimation, which is an advantage when measuring this variable involves complex instrumentation. Although this method does not achieve the same level of accuracy as other approaches. Furthermore, the computational cost is high because it is an iterative process and in each iteration it is necessary to perform a numerical integration. However, the results demonstrate that it is possible to obtain transfer functions representing the speed and current variables with an RMSE of less than 1%. Consequently, this method offers an alternative when access to two signals typically used as input data is not feasible. Otherwise, it is recommended to employ one of the heuristic methods extensively developed in the literature.

## Figures and Tables

**Figure 1 biomimetics-10-00146-f001:**
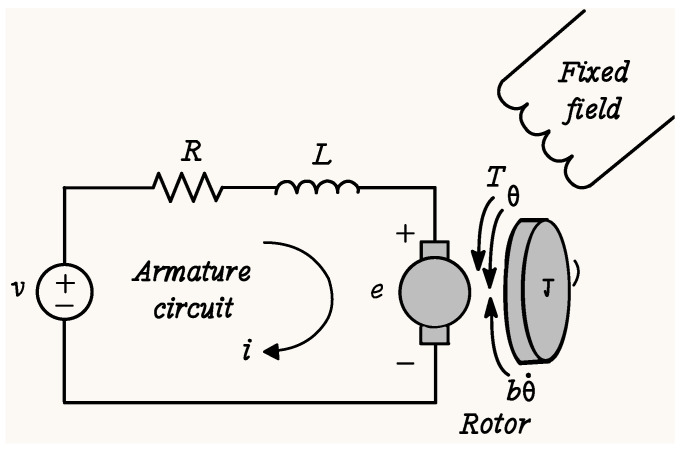
Diagram of a direct current motor.

**Figure 2 biomimetics-10-00146-f002:**
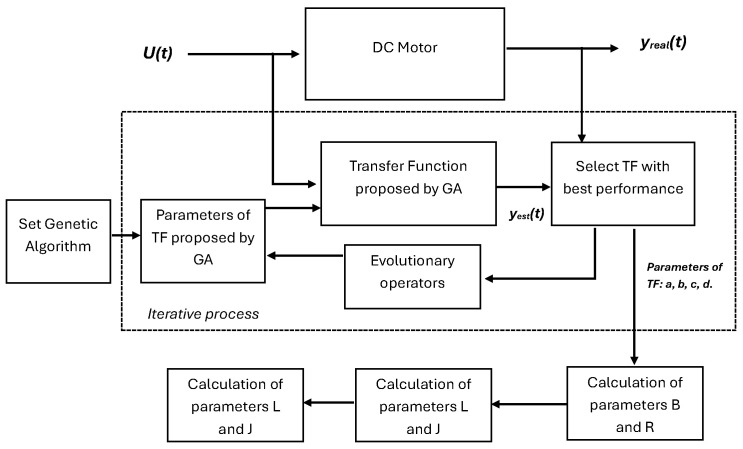
General outline of the steps used in this work to obtain the parameters of the dynamic model of the DC motor.

**Figure 3 biomimetics-10-00146-f003:**
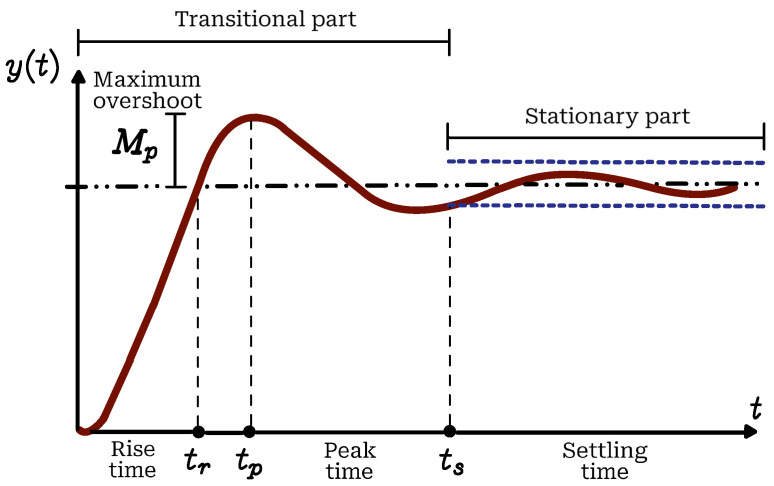
Typical dynamic response of a linear dynamic system to a step input.

**Figure 4 biomimetics-10-00146-f004:**
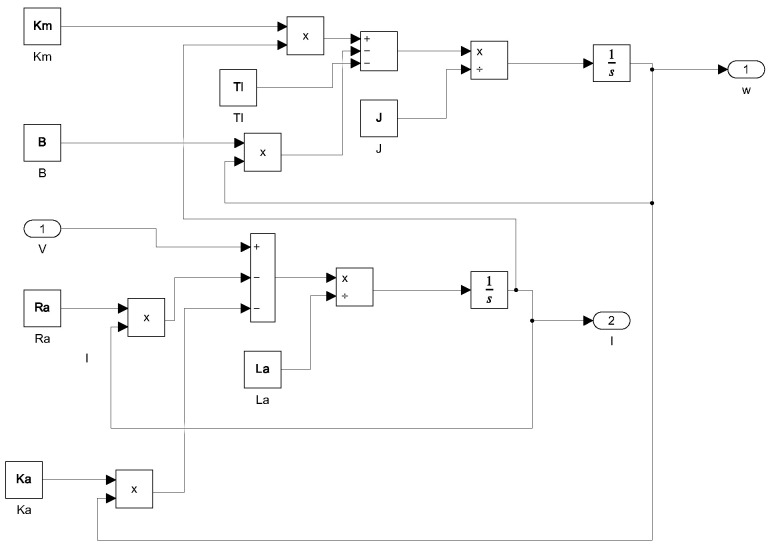
Simulink program to emulate the dynamical response of dc motors.

**Figure 5 biomimetics-10-00146-f005:**
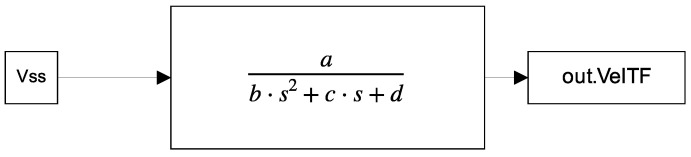
Simulink program to emulate the transfer function of velocity motors.

**Figure 6 biomimetics-10-00146-f006:**
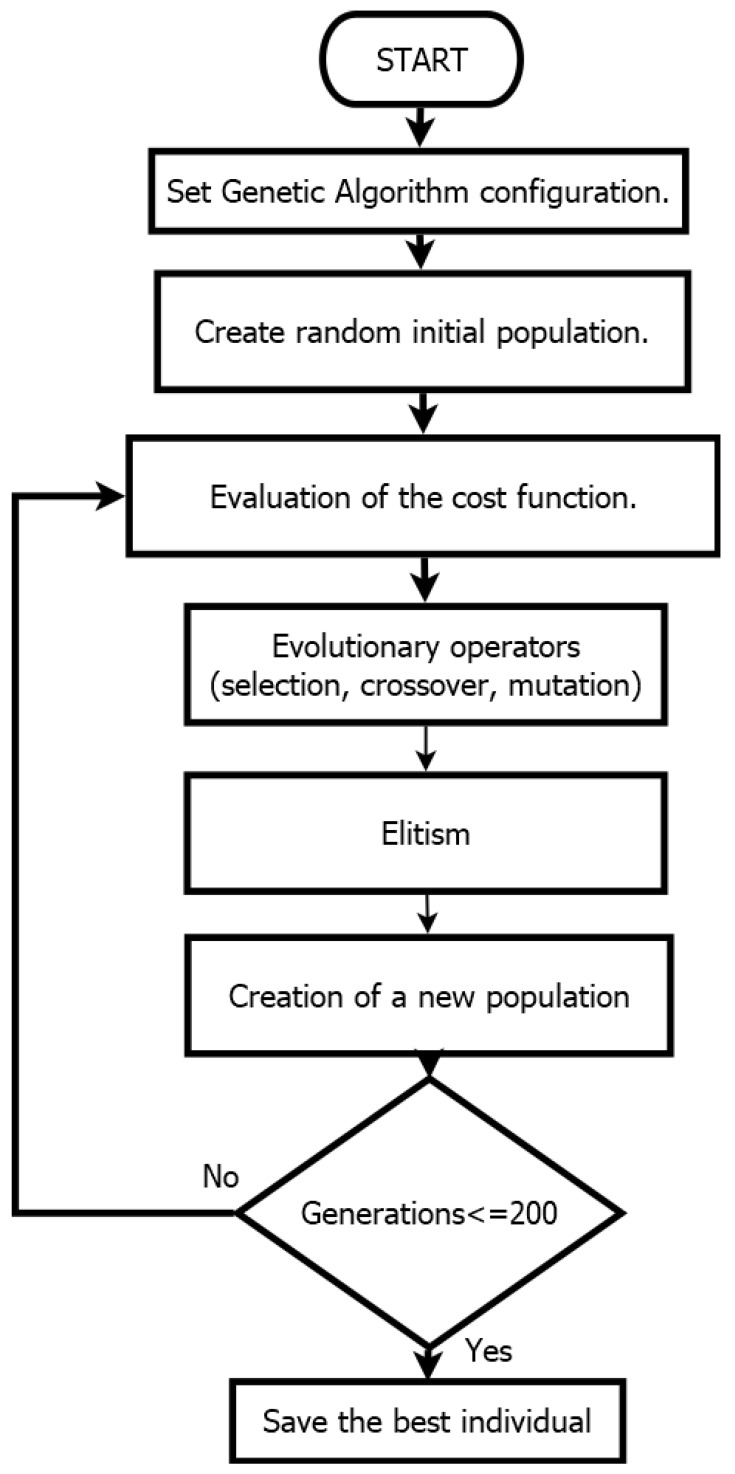
Flowchart of a genetic algorithm.

**Figure 7 biomimetics-10-00146-f007:**
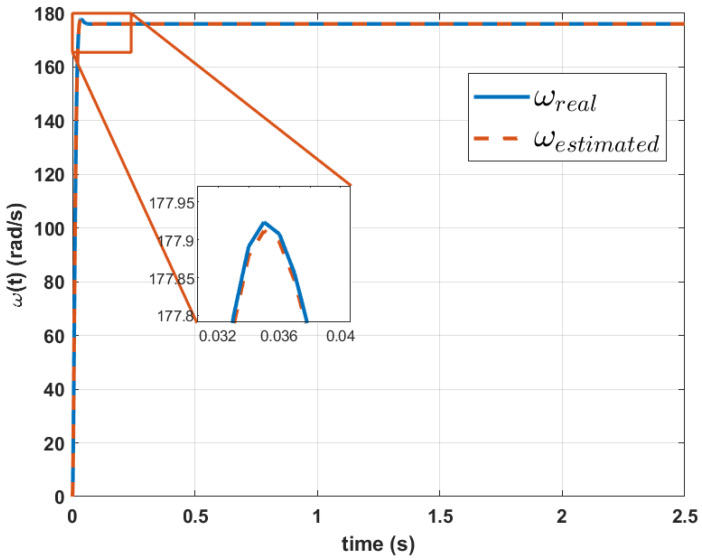
Comparison between the actual velocity signal and the estimated transfer function of velocity signal using Genetic Algorithms for Mavilor motor.

**Figure 8 biomimetics-10-00146-f008:**
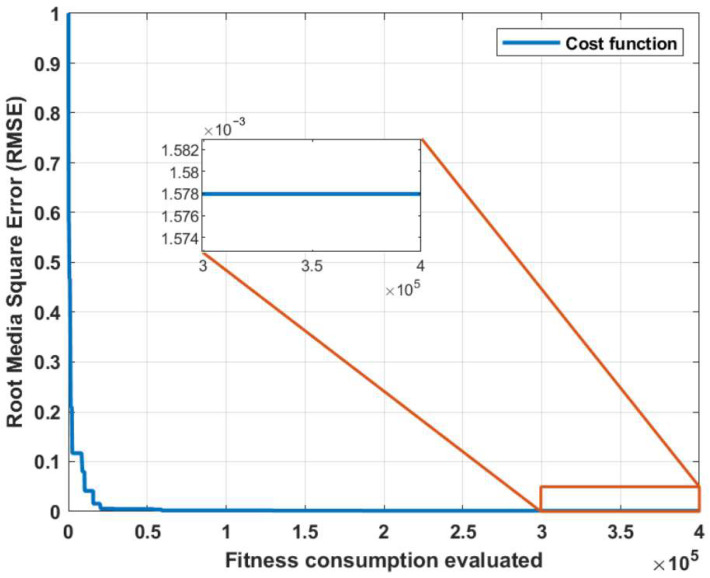
Cost function for estimating the parameters of the speed transfer function of the Mavilor motor.

**Figure 9 biomimetics-10-00146-f009:**
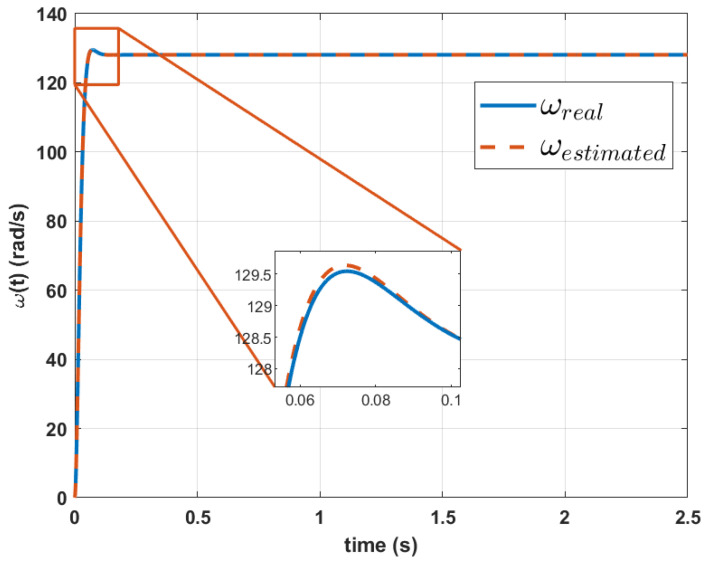
Comparison between the actual velocity signal and the estimated transfer function of velocity signal using Genetic Algorithms for RMCS2004 motor.

**Figure 10 biomimetics-10-00146-f010:**
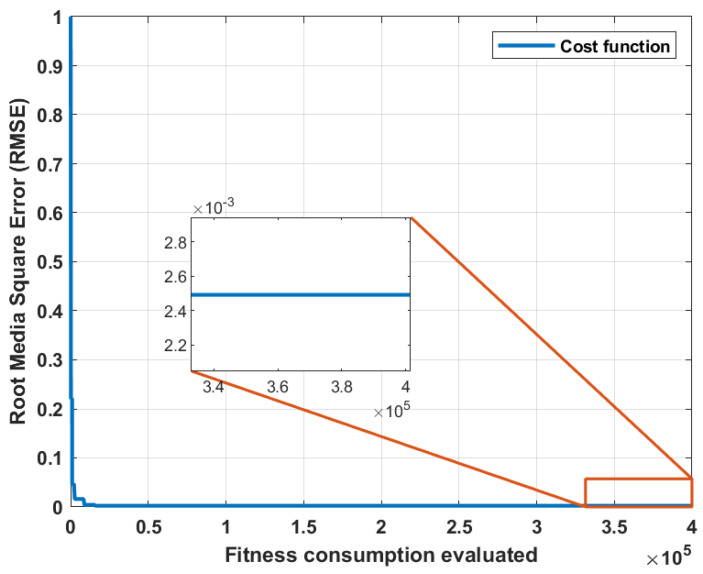
Cost function for estimating the parameters of the speed transfer function of the Mavilor motor.

**Figure 11 biomimetics-10-00146-f011:**
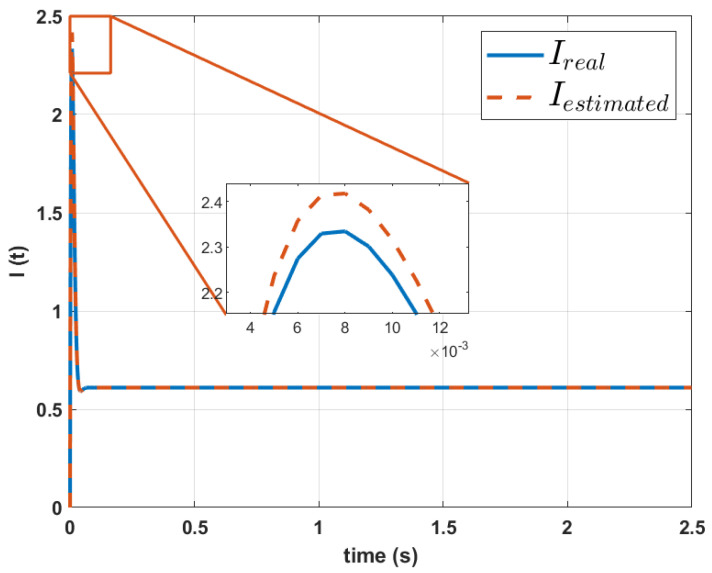
Comparison between the real current signal and the one calculated using the parameters of the Mavilor motor dynamic model estimated by the proposed method.

**Figure 12 biomimetics-10-00146-f012:**
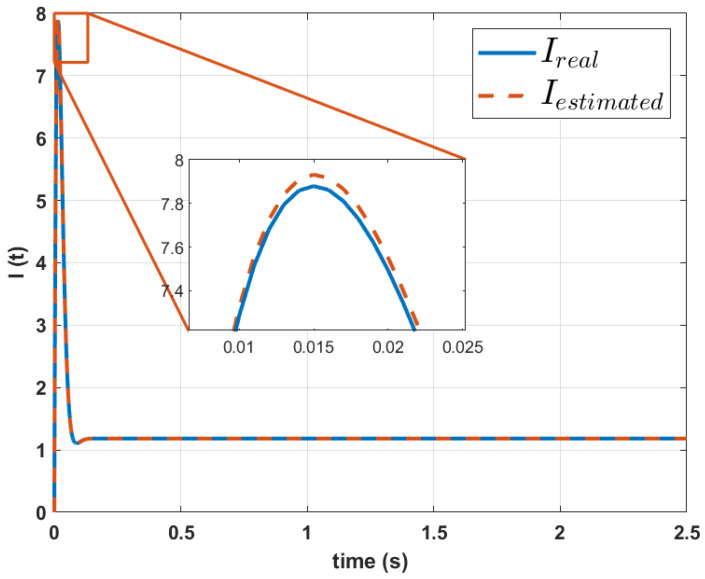
Comparison between the real current signal and the one calculated using the parameters of the RMCS motor dynamic model estimated by the proposed method.

**Table 1 biomimetics-10-00146-t001:** Methods for obtain parameters of a DC motor.

Work	Applied Method	Problem	Advantages	Limitations	Results
This work	Genetic Algorithms.	Parameterization of motor dynamic models using only speed signals.	Does not require multiple signals; less than 1% error in reconstruction; innovative and efficient approach.	Dependent on accurate input data; limited validation for specific systems.	Reconstruction of dynamic equations with RMSE < 1% for speed and current signals.
[23]	Particle Swarm Optimization (PSO) Algorithm.	Optimization of control parameters in robotic systems involving motors.	High precision in optimization; easy implementation.	Dependence on initial configuration; sensitivity to overfitting.	Parameter optimization with a 15% improvement in precision compared to traditional methods.
[24]	Deep Neural Networks (DNN)	Fault prediction in electrical systems, including motor components.	Advanced prediction capabilities; adaptable to non-linear data.	High computational cost; requires large datasets.	94% accuracy in predicting electrical faults.
[25]	GA	Parameter identification in mechanical systems, particularly motors.	Precise results for complex models; robust to noise.	Slow for large-scale problems; requires parameter tuning.	Accurate parameter identification with an average error below 2%.
[26]	Dynamic Modeling and Statistical Analysis.	Analysis of dynamic behavior in industrial systems, including motor dynamics.	Detailed modeling; enables precise analysis of complex dynamics.	Limited generalizability to other systems.	Dynamic models with a 90% accuracy rate in simulations.

**Table 2 biomimetics-10-00146-t002:** Nominal parameters of DC motor used.

Parameter (Units)	CML050 Nominal Value	RMCS2004 Nominal Value
K	0.048774	0.073472
B ($\frac{{\mathrm{kgm}}^{3}}{s^{2}}$)	0.000169	0.000678
R (Ω)	3.1363	0.921042
L (H)	0.01307	0.000136
J (Nm)	0.000009	0.000678

**Table 3 biomimetics-10-00146-t003:** Hiperparameters employed in Genetic Algorithm used as velocity signal estimator.

GA Hyperparameters	Value	Details
Population	2000	Numbers of vectors with random coefficients for a, b, c and d.
Upper search limit	[$\frac{V_{ss}}{\omega_{ss}}$ 1$\;\times \;10^{-6}$ 1$\;\times \;10^{-4}$ $\frac{V_{ss}}{I_{ss}}$]	Maximum value in the search for parameters.
Lower search limit	[$\frac{V_{ss}}{10\omega_{ss}}$ 1$\;\times \;10^{-8}$ 1$\;\times \;10^{-6}$ $\frac{V_{ss}}{10I_{ss}}$]	Minimum value in the search for parameters.
Fitness function	$RMSE=\sqrt{\frac{\sum_{i=0}^{2}\varepsilon^{2}dt}{n}}$	Function for evaluate the performance of each individual (Root Media Square Error)
Stop condition	genetration <= 200	Iterations that must be reached to stop the genetic algorithm
Elitism	1%	Percentage of the best individuals with guaranteed reproduction
Biological pressure	70%	Percentage of individuals that can reproduce
Mutation	30%	Probability of each individual suffering a mutation

**Table 4 biomimetics-10-00146-t004:** Comparison Nominal parameters of the Transfer Function of velocity against estimated parameters.

Parameter	CML050 Nominal Value	CML050 GA Value	RMCS2004 Nominal Value	RMCS2004 GA Value
a	4.8774 $\;\times \;10^{-2}$	4.92 $\;\times \;10^{-2}$	7.3472$\;\times \;10^{-2}$	7.354$\;\times \;10^{-2}$
b	1.1763$\;\times \;10^{-7}$	1.1893$\;\times \;10^{-7}$	1.0552$\;\times \;10^{-6}$	1.0609$\;\times \;10^{-6}$
c	3.0436$\;\times \;10^{-5}$	3.0777$\;\times \;10^{-5}$	1.3052$\;\times \;10^{-4}$	1.3032$\;\times \;10^{-4}$
d	2.908$\;\times \;10^{-3}$	2.949$\;\times \;10^{-3}$	6.022$\;\times \;10^{-3}$	6.028$\;\times \;10^{-3}$

**Table 5 biomimetics-10-00146-t005:** Comparison Nominal parameters of the dynamic model against estimated parameters.

Parameter (Units)	CML050 Nominal Value	CML050 GA Value	RMCS2004 Nominal Value	RMCS2004 GA Value
K	0.048774	0.0492	0.073472	0.0735
B ($\frac{Ns}{m}$)	1.69$\;\times \;10^{-4}$	1.7031$\;\times \;10^{-4}$	6.78$\;\times \;10^{-4}$	6.7868$\;\times \;10^{-4}$
R (Ω)	3.1363	3.0278	0.921042	0.9131
L (H)	0.01307	0.0126	0.007759	0.007745
J (Nm)	9.0$\;\times \;10^{-6}$	9.4573$\;\times \;10^{-6}$	1.36$\;\times \;10^{-4}$	1.3697$\;\times \;10^{-4}$

## Data Availability

Data are available upon reasonable request.

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
