# Peer review of "Model Parametrization-Based Genetic Algorithms Using Velocity Signal and Steady State of the Dynamic Response of a Motor"

_biomimetics, 2025, doi:10.3390/biomimetics10030146_

Round 1
Reviewer 1 Report
Comments and Suggestions for Authors
The paper is titled "Model Parametrization-Based Genetic Algorithms Using Velocity Signal and Steady State of the Dynamic Response of a Motor" and it is aimed to propose a novel method for motor parameter estimation by reconstructing the mechanical and electrical equations of a motor's dynamic model using only the velocity signal and steady-state values. This approach avoids reliance on complete current signal measurements and applies genetic algorithms for parameterization, achieving a robust estimation method suitable for resource-limited scenarios.
The topic is interesting and fits in the scope of the Biomimetics journal due to its content relation with "biologically inspired designs in engineering systems".
The references in the paper are well-formatted and adhere to the expected citation style, reflecting a professional presentation. Additionally, the references are up-to-date and highly relevant to the topic, demonstrating a comprehensive review of the existing literature.
The language used throughout the paper is clear and coherent, making it easy to follow and accessible to a broad academic audience. The writing style enhances the overall readability of the manuscript.
The abstract is well-constructed and effectively summarizes the key aspects of the study, providing a concise overview of the research objectives, methods, and findings.
However, I have identified several major issues that need to be addressed before the manuscript can be considered for publication. These comments pertain to critical areas requiring significant revision to enhance the quality and impact of the work.
1. The manuscript lacks sufficient details about the structure and implementation of the proposed Genetic Algorithm (GA) system. Specifically, the construction of the chromosome in the model is not clearly explained. From the current description, it is understood that the chromosome consists of four parameters: a, b, c, and d. However, it remains unclear whether these are the only variables or if additional variables/genes are included in the chromosome representation. This information is critical for evaluating the completeness and robustness of the proposed GA system.
2. How do you implement the Mutation operation in the proposed model?
3. How do you use the Crossoveroperation (what is your crossover operator) ?
4. Elitism is commonly employed in Genetic Algorithms to preserve the best solution(s) across generations, ensuring that high-quality solutions are not lost during the evolutionary process. However, the use of 10% elitism, as stated in the manuscript, appears to be disproportionately high. Such a large elitism percentage risks significantly reducing the diversity of the population, potentially leading to premature convergence and suboptimal solutions. To justify the use of this high elitism rate, do you have any specific metrics, experimental evidence, or problem-specific considerations supporting the decision to use 10% elitism?
5. There is an inconsistency between the stopping condition described in "Table 3: Hyperparameters employed in the Genetic Algorithm used as velocity signal estimator" and the one illustrated in "Figure 6: Flowchart of a Genetic Algorithm". In Table 3, the stopping condition is stated as generation >= 100, while in Figure 6, it is described as RMSE < 0.001. This discrepancy creates confusion about the actual termination criteria used in the algorithm. Providing a detailed explanation in the manuscript will enhance the clarity and reliability of the described methodology.
6. The Genetic Algorithm (GA) is an iterative optimization algorithm designed to enhance solutions over successive generations. However, the manuscript does not provide a visual representation of this iterative improvement, which is a key aspect of understanding the algorithm's performance. To better illustrate the effectiveness of the proposed GA, I recommend including a figure that depicts the improvement of the objective function (e.g., fitness score, RMSE, or other relevant metrics) across iterations or generations. This could be presented as a line graph, showing the progression of the best, worst, and/or average fitness values in the population over time. Such a figure would provide valuable insights into the convergence behavior, the rate of improvement, and the stability of the algorithm, thereby making the results more comprehensive and interpretable for readers.
7. In Figures 8 and 10, titled "Cost function for estimating the parameters of the speed transfer function of the Mavilor motor," the ordinate is labeled as "Root Media Square Error (ISE)." Could you clarify what this represents and how it is calculated? The paper does not provide any explanation or details about this term.
8. In the conclusion section, the authors state, "However, the results demonstrate that it is possible to obtain transfer functions representing the speed and current variables with an RMSE of less than 1%." However, earlier in the paper, the stated aim is to achieve "RMSE < 0.001." This discrepancy indicates that the stated objective has not been met. Could the authors clarify this inconsistency?
9. The content presented in "Table 1: Comparison of results in similar works" does not align with the title's implication of a comparative analysis. The table appears to focus solely on the application of Genetic Algorithms (GA) in the proposed work without presenting any actual comparative results with other studies. To ensure consistency and clarity, the table should either include relevant comparative data from similar works to justify its title or be retitled to accurately reflect its current content.
10. The explanation provided in Figure 6: Flowchart of a Genetic Algorithm lacks clarity, particularly regarding the breaking condition for the loop. It is mentioned that the condition is RMSE < 0.001, but it is unclear whether this condition applies to all chromosomes in the population or only to the best-performing chromosome. This distinction is crucial for understanding the stopping criteria of the algorithm and its implementation. I recommend elaborating on this aspect in the figure's description or the accompanying text to avoid ambiguity and provide a more comprehensive understanding of the algorithm's functionality.
Reviewer 2 Report
Comments and Suggestions for Authors
Dear Authors,
My main concern about the content of this work is:
1) What exactly is the contribution of this work? You make the following affirmation: "it has even been excitedly applied to parametric estimation of DC motor [27]." So, what is the difference between your work and [27]?
Other concerns:
2) There are several genetic algorithms in the literature. Provide a reference and mathematical explanation of the algorithm that you used;
3) The current estimation in Fig. 11 has a relevant overshoot error. I suggest running the GA for more time, ensuring that this algorithm finds the global solution.
4) Although the current estimation in Fig. 12 is better, it still has a relevant overshoot difference.
5) To evaluate fairly the obtained transfer functions, you must make reference changes during the experiment. In this way, you can evaluate the accuracy of obtained estimations in transient regimes. As the start-up has relevant errors, the same behavior will probably appear in transient regimes.
6) Compare the proposed method with other meta-heuristic algorithms.
Reviewer 3 Report
Comments and Suggestions for Authors
In this paper, the authors proposed an innovative approach to the parametrization of a motor's dynamic model using only the velocity signal and steady-state values, leveraging genetic algorithms (GAs) for parameter estimation. They argued that traditional methods often require multiple signals, complicating the modeling process. Although the paper is interesting, the following comments should be addressed.
-- In the abstract, clearly state the primary motivation for the research and the significance of using a single velocity signal.
-- In the introduction, expand on the importance of accurate modeling in various engineering applications, perhaps citing specific challenges.
-- At the end of the Introduction section, right before the last paragraph, the authors should indicate the major contributions of this paper by using 3 to 5 brief bullet points.
-- The structure of arguments has to be improved. You should have a section plan at the end of the introduction part, for example, section 2 discusses... and section 3 gives...
-- Enrich the literature review by discussing more recent publications to highlight the advancements in the field including the following ones: Evaluating the influence of AI on market values in finance: distinguishing between authentic growth and speculative hype; Resilient reinforcement learning for voltage control in an islanded DC microgrid integrating data-driven piezoelectric.
-- It is suggested to include more details on the experimental setup, such as the specifications of the motors used and the procedures followed during testing.
-- It is recommended to expand on the implications of the findings for future motor control strategies and applications in the industry.
-- It would be valuable to address potential limitations of the study in more depth, including any assumptions made during the modeling process.
-- If applicable, extend the methodology to different types of motors (e.g., stepper motors, brushless DC motors) to validate the robustness of the approach.
-- If possible, combine the parametrization approach with advanced control techniques (e.g., adaptive control, machine learning) to improve performance in dynamic environments.
Round 2
Reviewer 1 Report
Comments and Suggestions for Authors
The authors made related corrections.
It can be accepted as is.
Reviewer 2 Report
Comments and Suggestions for Authors
Dear Authors,
Thank you for improving the manuscript with my suggestions. I have no more concerns. Good luck with the publication!
Reviewer 3 Report
Comments and Suggestions for Authors
The authors have addressed all previously raised concerns in a satisfactory manner, demonstrating a clear understanding of the critical feedback and applying it constructively to enhance both the quality and clarity of the manuscript. In light of these substantial improvements and the rigorous effort expended by the authors to revise the paper, I recommend acceptance of the manuscript for publication.